# Autophagy and Polyphenols in Osteoarthritis: A Focus on Epigenetic Regulation

**DOI:** 10.3390/ijms23010421

**Published:** 2021-12-31

**Authors:** Consuelo Arias, Luis A. Salazar

**Affiliations:** 1Center of Molecular Biology and Pharmacogenetics, Department of Basic Sciences, Faculty of Medicine, Universidad de La Frontera, Temuco 4811230, Chile; consuelo.arias@ufrontera.cl; 2Department of Preclinical Sciences, Faculty of Medicine, Universidad de La Frontera, Temuco 4811230, Chile; 3Interuniversity Center for Healthy Aging (ICHA), Universidad de La Frontera, Temuco 4811230, Chile

**Keywords:** autophagy, osteoarthritis, aging, polyphenols, epigenetics, microRNAs

## Abstract

Autophagy is an intracellular mechanism that maintains cellular homeostasis in different tissues. This process declines in cartilage due to aging, which is correlated with osteoarthritis (OA), a multifactorial and degenerative joint disease. Several studies show that microRNAs regulate different steps of autophagy but only a few of them participate in OA. Therefore, epigenetic modifications could represent a therapeutic opportunity during the development of OA. Besides, polyphenols are bioactive components with great potential to counteract diseases, which could reverse altered epigenetic regulation and modify autophagy in cartilage. This review aims to analyze epigenetic mechanisms that are currently associated with autophagy in OA, and to evaluate whether polyphenols are used to reverse the epigenetic alterations generated by aging in the autophagy pathway.

## 1. Autophagy

Cellular homeostasis is dependent on intracellular mechanisms that maintain organelles and functional macromolecules that are required for cell survival and normal biosynthetic function [1]. Autophagy is an evolutionarily conserved homoeostatic process [2], initially described by Deter et al. in 1960 [3], and is characterized by being a highly conserved process by which cytoplasmic components (cytosolic macromolecules and dysfunctional organelles) are generally marked by ubiquitination and delivered to lysosomes for degradation and recycling [4,5,6]. This serves as a quality control mechanism as well as a recycling pathway [2]. There are at least three types of autophagy in eukaryotic cells: macroautophagy, microautophagy, and chaperone-mediated autophagy (CMA), differing between them in relation to the mode of delivery of the degradation contents to the lysosome [7,8,9]. Autophagy has a controversial role because it plays a dual function in both cell survival and apoptosis, since it can be used as a protection and a death mechanism in stressed cells [10,11,12,13], and it is suggested that the type of response depends on the cellular context [10,11]. Of all the known forms of cell death, including apoptosis, necrosis, and pyroptosis, among others [13], autophagy is described as a programmed non-apoptotic cell death [14]. The role of autophagy in facilitating cell death is very important, since it can remove senescent cells in aging tissues in addition to controlling the growth of neoplastic lesions [13].

In general, classic autophagy consists of a series of dynamic membrane arrangements mediated by a group of Autophagy Related Proteins (ATG) [1,6]. First, cytoplasm sequestration is generated within double-membrane vesicles called autophagosomes [5,6,9]. ULK1 is an autophagy inducer [9,10,11], and forms the first complex of this pathway with ATG13, FIP200 and ATG101 [2]. Beclin1 (BECN1), acts as an autophagy regulator [9,10,11] and, in conjunction with phosphatidylinositol 3-kinase catalytic subunit type 3 (PIK3C3) and phosphoinositide-3-kinase regulatory subunit 4 (PI3KR4), promotes engulfment and endosome maturation [15]. ULK complex is mobilized to the endoplasmic reticulum favoring the formation of the phagophore and allowing the recruitment of WIPI2B that recruits the E3-like complex ATG12– ATG5–ATG16L1 [16]. This last complex promotes the conjugation of LC3 [17]. LC3 is a critical biomarker of autophagosome formation and expansion and, when conjugated with phosphatidylethanolamine, results in the formation of another autophagy marker LC3-II [9,10,11]. Subsequently, the autophagosome fuses with the lysosome to generate autolysosomes guided by SNARE proteins [16]. Finally, the intracellular components are rapidly degraded by autophagolysosomal hydrolytic enzymes [17] and the constituents are released for biosynthesis or for their use as an energy source [5,6,9] (Figure 1). 

Autophagy pathways are integrated into multiple-signal transduction pathways that respond to the amount of available nutrients, energy balance, different cytokines, and growth factors [9]. A vital autophagy regulator is the interaction between serine and threonine kinases of the mammalian target of rapamycin (mTOR) in the mTOR 1 complex (mTORC-1) [18]. It functions as a central regulatory protein that integrates signals originating from intracellular and extracellular changes [19]. Inhibition of mTORC-1 promotes autophagy and activation of mTOR kinase suppresses autophagy [6,20]. Activation of extracellular signal-regulated kinase 1/2 (ERK1/2), protein kinase B (AKT), and p90 ribosomal S6 kinase (RSK1) promotes the activity of mTORC1 [19]. In fact, ERK1/2 signal cascade is activated in several autophagy models, and inhibition of this pathway inhibits autophagy [21]. Adenylate-activated protein kinase (AMPK) is a serine/threonine protein kinase that is activated through a combination of multiple phosphorylation by upstream kinases. AMPK is able to promote autophagy by acting differentially at different levels of autophagy regulation, for example by inhibition of mTORC-1 and phosphorylation of ULK1 [17,22]. Pathways of mTOR and AMPK are also linked to Sirtuin 1 (SIRT1) [23]. SIRT1 is a member of the class III histone deacetylase (HDAC) family and interacts with proteins linked to the regulation of autophagy such as ATG5, ATG7, LC3, Forkhead box O transcription factors (FoxO), E2F transcription factor 1 (E2F1), tumor protein p73 (TP73), PPAR-γ co-activator 1α (PGC1α), NF-κB and tumor protein p53 (TP53) [23]. Overexpression and activation of SIRT1 by resveratrol induces protective autophagy in non-small-cell lung cancer cells (NSCLC) via inhibiting Akt/mTOR and activating p38-MAPK pathway [24]. Even this activation of SIRT1 has been shown to be capable of inducing autophagy in enucleated cells [6]. Further, the reduction in the availability of nutrients generates the activation of sensors such as SIRT1, AMPK and mTOR, which increases the demand for autophagic replacement [4]. Finally, nuclear factor κB (NF-κB) is a transcriptional factor and is an important inducer of several autophagy related genes such as BECN1, BCL2 and sequestosome 1 (SQSTM1) [25]. It has been observed in degenerative human nucleus pulposus cells that inhibition of NF-κB blocks apoptosis and inflammation by promoting autophagy through AKT/mTOR pathway [26].

## 2. Autophagy and Aging

Autophagy is generating great interest due to its role in several physiological processes that are important for health and in age-related degenerative diseases [10,27].

Aging is frequently accompanied by defects in general autophagy, which causes a decrease in the ability of organisms to adapt to stress [5] and the accumulation of metabolic wastes, typical of cellular aging [28]. These effects have been observed in different aged rat tissues such as kidney, heart, cartilage, brain, and skeletal muscle, in which a significant decrease in the expression of LC3, BECN1 and ULK1 has been reported [29]. This decline in autophagic capacity in aging cells alters the cell maintenance process, promoting ROS generation and oxidative stress [22]. ROS accumulation and mTORC1 activation, are associated with accelerated aging and the development of age-related pathologies [1]. Some of these age-related diseases are characterized by the accumulation of autophagic vacuoles, which supports that the autophagic process is progressively altered with aging [30]. Finally, defects in autophagy can aggravate age-related alterations in model organisms, while the activation of autophagy protects against diseases related to aging and could lengthen life and reduce the severity of the disease [9,25].

In cartilage, autophagy is also considered a protective mechanism, for maintaining homeostasis and for being a cellular response to different types of stress [9,10,26,30]. It is thought that the development of structural changes in cartilage due to aging is linked to the alteration in homeostatic mechanisms such as autophagy [25,28] and that defects in autophagy cause tissue degeneration similar to that associated with aging [9,25]. It has been postulated that one of the connections could be the FoxO protein, given that FoxO controls: chondrocyte proliferation, cell viability by coordinating key cellular stress responses, articular cartilage homeostasis during aging and its overexpression significantly increased autophagic genes [31].

## 3. Autophagy and Osteoarthritis 

Osteoarthritis (OA) is a complex and multifactorial degenerative joint disease, it is one of the main causes of pain and dysfunction worldwide [29,32,33,34]. It is characterized by a degradation of the articular cartilage and a concomitant adaptive osteogenesis [30,35]. Although the cartilage has the most notable changes, the entire joint is affected, including the synovium, joint ligaments, and subchondral bone and it has been observed that inflammation from both synovitis and systemic inflammation play an important role in the genesis of this disease [31,36]. The chondrocyte itself contributes to joint degradation through enzymatic degradation of the extracellular matrix (ECM) (chondrocytic chondrolysis) [35]. Some of the changes that can be observed are: vascular infiltration, osteophyte formation, activation of macrophage, hypertrophic chondrocytes, fibrotic synovium, sclerotic bone formation, etc. [37].

Despite the fact that aging is one of the most important risk factors for OA [26,29,31,32,33,34,38], osteoarthritis is probably not a direct consequence of this, but it is aging itself that affects the ability of articular cartilage to maintain homeostasis such as autophagy [39]. Both aged human and mouse cartilage shows a reduction in autophagic protein expression [33]. By observing the formation of autophagic vesicles in cartilage as a measure of autophagic activity, Caramés et al. (2015) could demonstrate that there is a significant reduction in the level of basal autophagy in aged mice compared to young mice, and also mention that with aging there is a decrease in the expression of ATG5 and LC3 and that structural damage progresses in an age-dependent manner subsequent to changes in autophagy expression [33]. Also a reduction and loss of ULK1, BECN1, and LC3 has been associated with an increase in chondrocyte apoptosis in OA [1,11]. The defects in autophagy regulation in chondrocytes and aged cartilage have also been observed in OA models [40].

In OA a functional relationship between autophagy and apoptosis is also described, where in early stages of OA autophagy would be active to protect chondrocytes [14,41], while in later stages, autophagy could be active together with the apoptosis as an alternative pathway to cellular demise and could even induce senescence [14,42]. In this sense, it has been indicated that in chondrocytes and cartilage with OA versus healthy patients, autophagy may be increased with increased expression of autophagic markers. This increase is thought to be an adaptive response to protect cells from stress and regulate changes in OA-related gene expression through modulation of apoptosis and ROS during the phase initial degenerative OA, but in cases of severe damage autophagy would decrease [37,43]. In addition, it has been observed that the inflammatory stimulus with IL-1β in chondrocytes increases the LC3I protein expression similar to the one that was observed with the autophagy inducer, rapamycin, and this excessive or prolonged activation could promote cell death [44], these results are consistent with other studies [39,40,41].

On the other hand, it has been described that deletion of autophagy proteins could influence the development of OA. KO mice for some FoxO proteins generate alterations similar to those observed in cartilage due to OA and the overexpression of Fox1 reduces inflammatory mediators in chondrocytes with OA [31]. Since mTOR has been described to be overexpressed in human cartilage with OA, its deletion has been reported to favor protection against destabilization of medial meniscus (DMM)-induced OA [45].

There is no current cure for OA. Treatment can be classified into reduction of modifiable risk factors, intra-articular therapy, physical modalities, alternative therapies, and surgical treatments [46]. Exercise, patient education, and weight loss are the first line treatment recommended for knee and hip OA [47].

In the pharmacological aspect, there are no drugs available that are capable of modifying OA and there are a large number of drug candidates in clinical trials that have failed to demonstrate efficacy or are associated with adverse effects [1]. The most used pharmacological treatments in OA are generally symptomatic and focus on non-steroidal anti-inflammatory drugs (NSAIDs) and analgesics [44,48] and COX-2 inhibitors such as rofecoxib, that decrease pain but cannot stop disease progression, and intra-articular glucocorticoid and hyaluronic acid injections that decrease pain but may also increase disease progression [49]. For this reason, other therapeutic alternatives are necessary to prevent or decrease the progression of OA, avoiding the side effects of most treatments and focusing on the multimodal and progressive nature of OA [50].

## 4. Autophagy as a Therapeutic Target in OA 

There are currently several treatments for OA that promote autophagy. An example is rapamycin, an mTOR inhibitor, which induces autophagy in various cell types including in particular chondrocytes limiting joint damage [9,51]. Autophagy activation by rapamycin reduces intracellular levels of ROS induced by IL-1β and reduces cartilage destruction in experimental OA models [41]. Rapamycin is a useful drug for the study of OA in different models, but it can have potential adverse effects, such as disorders of the blood, metabolism, and nervous system, among others, which limits its use in long-term treatments [9]. Another drug that is safer for long-term administration and is commonly prescribed to relieve OA in humans is glucosamine [52]. Glucosamine prevents demethylation of IL-1β resulting in decreased expression [53], it also modulates targets of the autophagy pathway in vitro and in vivo in a manner dependent on the AKT/FoxO/mTOR pathway [9]. Some researchers have investigated other ways of regulating autophagy, for example through microRNAs. The miRNA-335-5p activates autophagy in human OA chondrocytes, increasing its viability and reducing its inflammatory mediators [54]. On the other hand, deregulation of miR-128a targeting ATG12, impairs chondrocyte autophagy and accelerates development of OA, and its interruption attenuated chondrocyte dysfunction and delayed OA development [55]. The development of safe and effective drugs that can enhance autophagic activity or restore autophagy flux is a promising strategy for the treatment of OA [42].

There are several components that have been suggested to have anti-senescence activity, among these are polyphenols, especially for their antioxidant and anti-inflammatory activity at the systemic level [45,56].

## 5. Polyphenols and Autophagy 

Polyphenols are the most common bioactive natural products [57], and are present in fruits, vegetables, seeds, and nuts [45,52,53]. The beneficial effects of polyphenols have been attributed to their antioxidant capacity and their ability to modify antioxidant cell defense mechanisms by inducing the synthesis of different enzymes [54,55], to its anti-inflammatory effect that inhibits chronic inflammation associated with aging [58], and several polyphenols can also affect numerous cell targets that have the ability to induce or inhibit autophagy [28].

Polyphenols are known to be able to activate autophagy through various mechanisms [59] and could regulate autophagy under different conditions. It has been reported that the use of polyphenols could prevent the side effects of the use of doxorubicin, a chemotherapeutic agent, which generate a dysregulation of autophagy and these effects would be through an induction of autophagy [60]. Another article mentions that the use of polyphenols in diabetic cardiomyopathy could improve metabolic disorders through a regulation of autophagy [61]. In the case of diseases associated with the nervous system such as Alzheimer’s and Parkinson’s, it is described that there is a dysregulation of autophagy and the use of polyphenols would increase markers associated with this pathway [8]. Moreover, the use of polyphenols could prevent oxidative stress, inflammation and inhibition of autophagy by aging in the brain [62].

The pharmacological role of polyphenols depends on their bioavailability, which differs among them due to their differences in chemical structure and their biotransformation until reaching the bloodstream [8].

## 6. Polyphenols used to Regulate Autophagy in OA 

Because autophagy is a key process for maintaining healthy cartilage and has been reported to decline with aging, several authors evaluated the effect of some polyphenols as a treatment for OA, targeting autophagy. The effects of some polyphenols on regulation of autophagy in OA are summarized in Figure 2 and Table 1. 

Resveratrol: In a model of OA due to destabilization of the medial meniscus, the use of intra-articular injections of Resveratrol (RV) delayed the degradation of articular cartilage, evaluated through the OARSI score system [63]. They also reported an increase in autophagy and col2a1 markers and a decrease in metalloproteinase 13 (MMP13) and ADAMTS5 in part by regulating HIF1α and HIF2α via AMPK/mTOR pathway [63]. Other authors indicate that the use of RV would block the decrease in autophagy through the AMK/SIRT1 pathway in cells of the nucleus pulposus stimulated with TNFα [64]. In this same cell type, stimulated with H_2_O_2_, RV treatment increases the expression of autophagy markers such as Beclin-1 and LC-3 and increases their activity through the PI3K/AKT pathway [65].

Butein: The use of butein (a polyphenol present in several plants) has been reported to increase autophagic flux by increasing phosphorylation of AMPK, TSC2 and ULK1 and inhibiting mTOR phosphorylation, in human cartilage with OA and in chondrocytes stimulated with IL1β. The authors postulate that butein activates autophagy of chondrocytes with OA through the AMPK/TSC2/ULK1/mTOR pathway [66]. 

Olive polyphenols: It has also been suggested that the use of olive polyphenols would increase autophagy through suirtin-1 signaling [67].

Mangiferin: The use of mangiferin (a natural polyphenol) in chondrocytes stimulated with tert-butyl hydroperoxide (TBHP) has been reported to increase the expression of autophagy markers such as LC3II/LC3I and ATG5 and increase autophagic flux and this increase depends on the activity of p-AMPK [43].

Curcumin: The use of curcumin retards aging-related cartilage degradation in mouse articular cartilage and enhances chondrocyte autophagy in knee joints of mice with surgically induced and age-related OA. This effect is via Akt/mTOR signaling pathway and contributes to the anti-OA effect of curcumin [68].

Propolis polyphenols: The use of polyphenols present in propolis in IL1β-stimulated chondrocytes regulates the expression of proteins associated with autophagy [44].This article mentions that this regulation is through a reduction in the expression of proteins associated with autophagy towards normal levels in unstimulated chondrocytes and the mechanism involved is through a reduction of oxidative stress that is generated by the application of the inflammatory stimulus [44].

## 7. Epigenetics and OA

During the development of OA both environmental and genetic factors influence chondrocyte biology through epigenetic regulation [69] and, therefore, these epigenetic modifications could represent a therapeutic opportunity for OA, a currently intractable disease [70].

Epigenetics is a study of hereditary changes excluding DNA sequence modifications that control gene expression and they have a decisive role in the patterns of physiological and pathophysiological processes. In contrast to genetic changes, epigenetic modifications are reversible [59]. Epigenetic mechanisms include DNA methylation, histone modifications and non-coding RNAs. 

DNA methylation is the most studied DNA modification in humans [52], it is the only epigenetic mechanism that directly affects the nucleotide bases of DNA and is important for numerous biological functions involved in its development [57]. It consists of the addition of a methyl group to the carbon in position 5 ’of a cytokine arranged in the cytosine residues of the dinucleotide sequence CpG, known as CpG islets, which in turn are located in promoter regions of regulatory genes, catalyzed by DNA methyl-transferase (DNMT) enzymes [64,65]. DNMTs can set the initial methylation patterns (de novo DNMTs) or maintain the already established methylation signature (maintenance DNMTs) [71]. Methylation is linked to repressed chromatin states and inhibition of the initiation of transcription, which is associated with a long-term shutdown of the associated gene [64,65,67].

The epigenetically conformation can be regulated also by modifications on the tails of histones which form the nucleosome protein core [72]. These modifications trigger chromatin remodeling, which determines the state of chromatin activity [52]. Histone modification includes methylation, acetylation, phosphorylation, ubiquitination, SUMOylation, and ADP-ribosylation, which can activate or suppress gene transcription by modifying the structure of chromatin [16,64]. Histone lysine acetyltransferase enzymes (HATs) catalyze the acetylation of histone lysine residues and deacetylase enzymes (HDACs) remove the acetyl group [70]. In general, histone acetylation is associated with genetic activation that promotes gene expression [45,63]. 

Lastly, noncoding RNAs (ncRNAs) include: microRNAs (miRs, 20-23 nt), snoRNAs, piwi interacting RNAs (piRNAs), and long ncRNAs (> 200 bp). These are able to mediate sequence-specific modulation of gene expression by different mechanisms [45,68]. The miRNAs are important regulators of gene expression and repress translation [45,64,69]. They are initially transcribed by RNA polymerase II (Pol II) in the nucleus from intragenic, intergenic or splicing regions to form long pri-miRNA transcripts [64,70]. These transcripts are processed by the enzymes RNAse III, Drosha, and Dicer to generate the mature miRNAs of 18–24 nucleotides [73]. The miRNAs bind to the three ’untranslated regions (UTR) of the target messenger RNAs (mRNA) [7] and function as negative regulators of gene transcription through two mechanisms [57]. The first involves the perfect binding of miRNA to the mRNA sequence, which results in the direct degradation of the target mRNA, preventing its translation [49,74]. The second mechanism involves incomplete binding, which represses the translation of the target mRNA [57].

## 8. Epigenetic Regulation of Autophagy 

Epigenetic regulations of autophagy are an essential mechanism for maintaining homeostasis and its dysregulation can generate different pathologies [75]. Hu in 2019, in the chapter of Epigenetic Regulation of Autophagy of Autophagy: Biology and Diseases, explores that many autophagy-related genes such as ULK Kinase/ATG1, Beclin1/ATG6, LC3/Atg8, and LAMP2 have been found to be methylated and silenced, thus inhibiting the process of autophagy and autophagic flow [19]. On the other hand, it also mentions that DNA methylation can also modify the genes that encode autophagy regulatory signal molecules such as: Nitro Domain-Containing Protein 1 (NOR1), Death-Associated Protein Kinase (DAPK), and SOX1 [19]. Other studies also analyze the relationship between methylation and autophagy. Increased ROS levels have been reported in gestational diabetes mellitus (GDM) model and this has caused a global DNA methylation by increasing DNA methyltransferase (DNMT3A). Increased DNMT3A attenuated cardiac Sirt 1 protein and p-Akt/Akt, raising autophagy-related proteins expression (Atg 5 and LC3 II/LC3 I) as compared to controls. This results in aberrant development of heart ischemia-sensitive phenotype in offspring [75]. Arginine methylation is an epigenetic modification involved in autophagy and is catalyzed by the protein arginine methyltransferase (PRMT) family [76]. In this study, the authors mention that coactivator-associated arginine methyltransferase 1 (CARM1) methylates Pontin chromatin-remodeling factor under glucose starvation and interacts with FoxO3, a transcription factor for transcriptional activation of autophagy genes [76]. Another study analyzes epigenetic memory of autophagy [77]. In this research they establish that there is a relationship between the stimulation of autophagy and the subsequent methylation of DNA by DNMT3A of MAP1LC3 which generates a prolonged decrease in the basal autophagy level through a persistent downregulation of MAP1LC3 [78].

It has been described that autophagy activation induced by several stimuli in different cell lines is related to a global reduction of H4K16 acetylation and in the case of H3K56ac, its acetylation could generate a positive regulation of the TOR signaling [19].This agrees with the report by Füllgrabe et al. (2013), where they relate a decrease in acetylation of histone H4 lysine 16 and transcriptional regulation of autophagy-related genes [79]. Another study indicates that overexpression of histone HIST1H1C, a variant of linker histone H1, upregulates SIRT1 and HDAC1 to maintain the deacetylation status of H4K16, leads to an upregulation of the ATG12–ATG5 complex, ATG7, ATG3, and LC3B-I to LC3B-II conversion, inducing autophagy in cultured retinal cell line [80].

In the case of miRNAs, it has been shown that they are involved in many stages of autophagy, including autophagic induction, vesicle nucleation, vesicle elongation, vesicle retrieval and fusion, and they are also implicated in the regulation of upstream signaling pathways that can affect autophagy induction [19]. Several microRNAs are known to inhibit autophagy by decreasing the expression of autophagy-related mRNAs. In breast cancer cells miR-101 is a strong inhibitor of basal and induced autophagy [81,82,83]. MiR-376b inhibits autophagy by acting on ATG4 and BECN1 while miR-630 inhibits autophagy by acting on ATG12 and UV radiation resistance associated gene protein (UVRAG) [84]. MiR-30a can negatively regulate autophagic activity by binding to BECN1 by decreasing its expression [7]. Other miRNAs enhance autophagy by acting on anti-autophagic molecules mRNAs. For example, miR-26a acts on negative regulators of autophagy such as induced myeloid leukemia cell differentiation protein (MCL1), TAK1-binding protein 2 (TAB2), Cytochrome c oxidase 5A (COX5A ), Polymerase (RNA) III (DNA directed) polypeptide G (POLR3G) and two negative regulators of MAPKs such as Dual specificity protein phosphatase 4 (DUSP4) and Dual specificity protein phosphatase 5 (DUSP5), resulting in activation of MAPKs and increased BECN1 expression [85]. The miR-325 suppresses the caspase recruitment domain (ARC) (anti-autophagic protein) which may increase autophagic activity in mice models [7]. Since B-Cell Leukemia/Lymphoma 2 (BCL-2) can bind BECN1 and inhibit BECN1-dependent autophagy, miRNAs that target BCL-2 such as miR-182, miR-34a, miR-210, miR-205 and miR-21 could regulate autophagy [7]. Figure 3 presents some examples of epigenetic regulation of autophagy.

## 9. Epigenetic Regulation of Autophagy in OA 

Despite the existence of plenty of information related to the different mechanisms of epigenetic regulation in OA [66,80,81], there is limited information relative to DNA methylation and histone modification regulation of autophagy in OA model. In relation to this, we found a study that indicates that ubiquitin-like with PHD and RING finger domains 1 (UHRF1), an epigenetic regulatory factor, could control DNA methylation, histone acetylation, and histone methylation involved in OA and autophagy [86].

It was described that a UHRF1 expression increase in human OA and a down-regulation of UHRF1 induce an increase in cell proliferation and autophagy through PI3K/AKT/mTOR signaling pathway [86].On the other hand, it was suggested that SIRT3 in chondrocytes could regulate mitophagy, a type of selective autophagy through the regulation of the deacetylation of FoxO1 and FoxO3a, essential transcription factors for the activation of the autophagy pathway [77].

In relation to miRs regulation of autophagy in OA, the literature is more abundant. Abnormal levels of miRs have been reported in chondrocytes during the onset of OA, the majority of which are functionally involved in the apoptosis and autophagy of chondrocytes at epigenetic, transcriptional, and post-transcriptional levels [87].Some autophagy-associated miRs that have been studied in OA are described in Table 2 and Figure 4. 

## 10. Epigenetics, Polyphenols, Autophagy and OA

In recent years, dietary polyphenols have shown to have the ability to regulate expression patterns of multiple genes through epigenetic modulation mechanisms [52,58]. It is suggested that polyphenols reverse altered epigenetic regulation by changing DNA methylation, histone modification, and modulating miRNA expression or directly interacting with enzymes [57] and because it is thought that, since epigenetic changes occur first [59], a strategy based on this could prevent certain age-related diseases such as cancer, neuroinflammation, diabetes, and aging [94]. In addition, polyphenols can be considered as a calorie restriction (CR) mimetic acting on the same signaling pathways as CR-regulating autophagy through epigenetic changes [59]. 

It is described that treatment with resveratrol (RV) induces epigenetic changes such as miR-1260a, miR-141-3p, miR-424-5p, miR-15a-5p, miR-7-5p and counteracts IL-6 migration of ovarian cancer cells through induction of autophagy [95]. Another study analyzed the effect of RV and 5-azacytydine treatment in adipose stem cell population (ASC). They observed that autophagy could be down-regulated to the basal level observed, improved the metabolic status with an up-regulation of miR-514a, and mentioned that RV could mediate this effect through dual regulation under different conditions [96].

Although there is a large amount of information on epigenetic regulation of the autophagy pathway, there are few studies that relate this information to OA and so far, no research has been found that mentions polyphenols’ epigenetic effect on the autophagy in osteoarthritis. The miRNAs have been shown to have important roles in several biological and pathological processes, while being able to regulate the expression of multiple genes. OA is a multifactorial pathology so therapeutic strategies based on microRNAs could affect several targets, surpassing conventional therapeutic strategies.

## 11. Future Perspectives 

Autophagy is a mechanism for maintaining cellular homeostasis in various tissues and is particularly important in cartilage. This process is dynamic and can be regulated by various factors. With aging, autophagy is deregulated, its rate decreases and therefore waste accumulates in cells, which further favors tissue aging [5]. In fact, senescent cartilage autophagy disorders are correlated with cell death and OA [97]. While the dysregulation of autophagy in OA has already been characterized, and some treatments such as polyphenols have been postulated to improve dysregulation, there are still concerns about therapeutic strategies that focus on this multi-factorial disease. It has already been hypothesized that with an unfavorable epigenetic profile there would be a tendency to generate or develop OA more rapidly [98].

Since epigenetic defects occur in the early stages of various diseases, interventional approaches targeting the epigenome have been proposed as preventive strategies [94]. Therefore, natural treatments such as polyphenols, that could maintain the healthy chondrocyte phenotype through epigenetic mechanisms that regulate autophagy, represent an attractive therapeutic strategy, taking into account the reversible nature of these epigenetic alterations. 

## Figures and Tables

**Figure 1 ijms-23-00421-f001:**
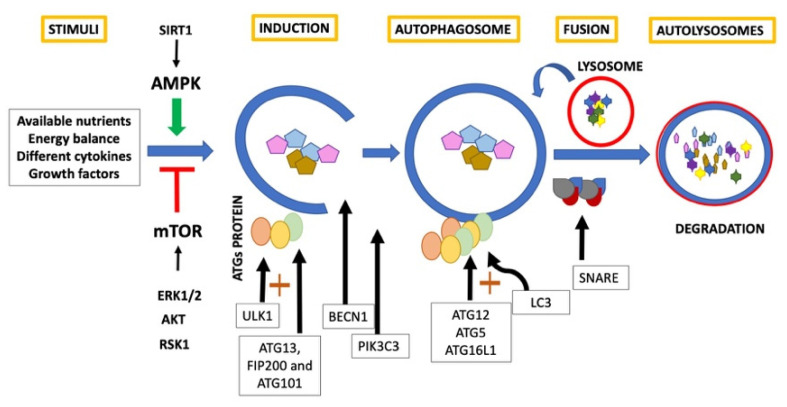
Classic autophagy.

**Figure 2 ijms-23-00421-f002:**
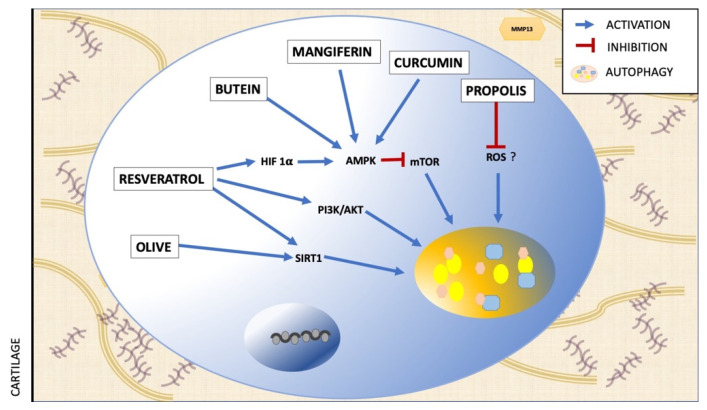
Polyphenols used to regulate autophagy in OA.

**Figure 3 ijms-23-00421-f003:**
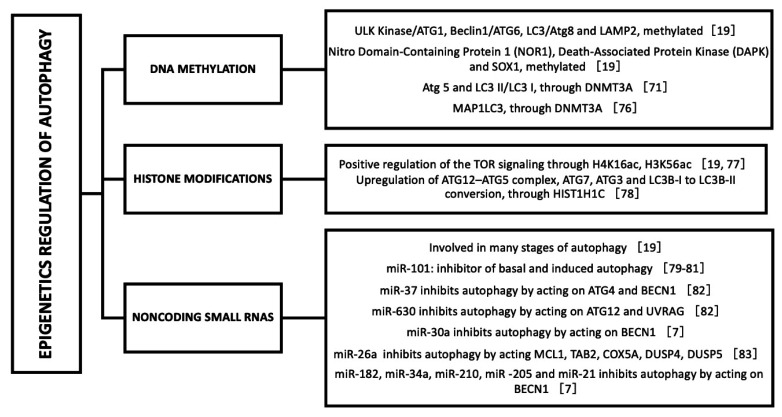
Epigenetic mechanisms involved in autophagy.

**Figure 4 ijms-23-00421-f004:**
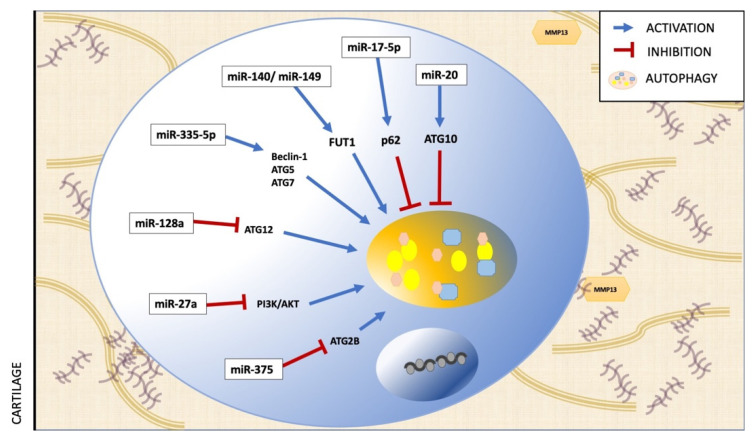
Regulation of autophagy by miRs in osteoarthritis.

**Table 1 ijms-23-00421-t001:** Effects of polyphenols in autophagy in OA model.

Polyphenol	Assay	Effect on Autophagy	Mechanism	Ref.
Resveratrol	in vivo	Activation	AMPK/mTOR pathway	[63]
	in vitro	Activation	AMK/SIRT1 pathway	[64]
	in vitro	Activation	PI3K/AKT pathway	[65]
Butein	ex vivo	Increase of autophagic flux	AMPK/TSC2/ULK1/mTOR pathway	[66]
Olive polyphenols	in vitro	Activation	Suirtin-1 signaling	[67]
Mangiferin	in vitro	Increase of autophagic flux	AMPK signaling	[43]
Curcumin	in vitro	Activation	Akt/mTOR signaling pathway	[68]
Propolis polyphenols	in vitro	Reduction of protein autophagy	ROS signaling	[44]

**Table 2 ijms-23-00421-t002:** Regulation of autophagy by miRNAs in Osteoarthritis (OA).

miRNA	Assay	Model/Cell Line	Target in Autophagy	Effect on Autophagy	Ref.
miR-335-5p	in vitro	Chondrocytes from Human OA articular cartilage	Beclin-1, ATG5, ATG7	Activation	[54]
miR-128a	in vitro in vivo	Chondrocytes from Human OA articular cartilage Rat anterior cruciate ligament transection (ACLT)	ATG12	Inhibition	[55]
miR-27a	in vitro	Chondrocytes from OA human articular cartilage and traumatic amputees.	PI3K	Inhibition	[88]
miR-375	in vitro in vivo	Chondrocytes from OA human articular cartilage. Destabilization of the medial meniscus	ATG2B	Inhibition	[89]
miR-140-5p/ miR-149	in vitro	Chondrocytes from OA human articular cartilage and normal donors.	FUT1	Inhibition	[90]
miR-17-5p	in vitro in vivo	SW1353 human chondrosarcoma cells Destabilization of the medial meniscus on C57BL/6J mice	p62	Activation	[91]
miR-20	in vitro in vivo	Chondrocytes from OA human articular cartilage and traumatic amputees. Sprague-Dawley (SD) rats	ATG10	Inhibition	[92]
miR-411	in vitro	Human chondrocyte C28/I2 line	Beclin-1, P62, ULK-1, LC3	Inhibition	[93]

## Data Availability

No new data were created or analyzed in this study. Data sharing is not applicable to this article.

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
