# Peer review of "Autophagy and Polyphenols in Osteoarthritis: A Focus on Epigenetic Regulation"

_ijms, 2021, doi:10.3390/ijms23010421_

Round 1
Reviewer 1 Report
In the manuscript “Autophagy and Polyphenols in Osteoarthritis: A Focus on Epigenetic Regulation” Arias and co-workers reviewed the role of polyphenols toward epigenetic regulation of autophagy in Osteoarthritis.
In my opinion this paper is interesting, however I suggest the following Minor changes in order to further improve it for publication in International Journal of Molecular Sciences:
1) In page 9, table 1, when referring to experiments that have been carried out both in cells and in animals (i.e. 54 reference), Authors should use the expression “in vitro” for cell cultures, while “in vivo” for animal studies.
2) There are also some grammatical, typing and punctuation errors, hence moderate English language and style editing is required.
Author Response
Response letter to reviewer comments (Manuscript ID: ijms-1506883)
Reviewer #1: Dear reviewer: We greatly appreciate the suggestions made. In order to address your observations, we incorporate your comments and the corresponding answer in this document, as detailed below.
Question 1: In page 9, table 1, when referring to experiments that have been carried out both in cells and in animals (i.e. 54 reference), Authors should use the expression “in vitro” for cell cultures, while “in vivo” for animal studies.
The table was corrected. We included the information requested. Please see Table 2
Question 2: There are also some grammatical, typing and punctuation errors, hence moderate English language and style editing is required.
The grammar and redaction were revised.

Reviewer 2 Report
Comments:
In this manuscript, the authors described “Autophagy and Polyphenols in Osteoarthritis: A Focus on Epigenetic Regulation.”. This review aims to analyze epigenetic mechanisms that are currently associated with autophagy in osteoarthritis, and to evaluate whether polyphenols are used to reverse the epigenetic alterations generated by aging in the autophagy pathway. However, there are a few points that need to be clarified.
Comment
- Apoptosis and autophagy are important molecular processes that maintain organismal and cellular homeostasis, respectively. The author will describe the correlation between apoptosis and autophagy in osteoarthritis.
- Several polyphenols can affect numerous cell targets that have the ability to induce or inhibit autophagy. I should draw a chart to illustrate what kind of polyphenols have the ability to induce or inhibit autophagy.
Author Response
Response letter to reviewer comments (Manuscript ID: ijms-1506883)
Reviewer #: Dear reviewer: We greatly appreciate the suggestions made. In order to address your observations, we incorporate your comments and the corresponding answer in this document, as detailed below.
Question 1. Apoptosis and autophagy are important molecular processes that maintain organismal and cellular homeostasis, respectively. The author will describe the correlation between apoptosis and autophagy in osteoarthritis.
We included the information requested. Please see pages 1 and 4.
Question 2. Several polyphenols can affect numerous cell targets that have the ability to induce or inhibit autophagy. I should draw a chart to illustrate what kind of polyphenols have the ability to induce or inhibit autophagy.
We included the information requested. Please see Table 1.

Reviewer 3 Report
This review is very interesting and well written in all its parts. This reviewer, thinks it can be accepted for pubblication after minor typos check:
line 38-40: acts and promotes
line 290: explores
Author Response
Response letter to reviewer comments (Manuscript ID: ijms-1506883)
Reviewer #3: Dear reviewer: We greatly appreciate the suggestions made. In order to address your observations, we incorporate your comments and the corresponding answer in this document, as detailed below.
This review is very interesting and well written in all its parts. This reviewer, thinks it can be accepted for publication after minor typos check:
Line 38-40: acts and promotes
Line 290: explores
The grammar was revised, and the typos checked.

Round 2
Reviewer 2 Report
accepted